# Peripheral optogenetic stimulation induces whisker movement and sensory perception in head-fixed mice

**Sunmee Park, Akhil Bandi, Christian R Lee, David J Margolis\***

Department of Cell Biology and Neuroscience, Rutgers, The State University of New Jersey, Piscataway, United States

**Abstract** We discovered that optical stimulation of the mystacial pad in Emx1-Cre;Ai27D transgenic mice induces whisker movements due to activation of ChR2 expressed in muscles controlling retraction and protraction. Using high-speed videography in anesthetized mice, we characterize the amplitude of whisker protractions evoked by varying the intensity, duration, and frequency of optogenetic stimulation. Recordings from primary somatosensory cortex (S1) in anesthetized mice indicated that optogenetic whisker pad stimulation evokes robust yet longer latency responses than mechanical whisker stimulation. In head-fixed mice trained to report optogenetic whisker pad stimulation, psychometric curves showed similar dependence on stimulus duration as evoked whisker movements and S1 activity. Furthermore, optogenetic stimulation of S1 in expert mice was sufficient to substitute for peripheral stimulation. We conclude that whisker protractions evoked by optogenetic activation of whisker pad muscles results in cortical activity and sensory perception, consistent with the coding of evoked whisker movements by reafferent sensory input.

**\*For correspondence:** david.
margolis@rutgers.edu

**Competing interests:** The authors declare that no competing interests exist.

## Introduction

Active sensing involves the integration of internally generated motor commands with sensation of the external world. In the rodent whisker system, which has been used extensively as an experimental model of active sensing, animals use their mystacial vibrissae (whiskers) to sample the immediate environment in rhythmic bouts of active self-generated whisker movements, called whisking (*Kleinfeld et al., 2006*; *Brecht, 2007*; *Diamond et al., 2008*; *Prescott et al., 2011*). A temporal sequence of extrinsic and intrinsic whisker pad muscle activation drives exploratory whisking: the extrinsic muscle *M. nasolabialis profundis* initiates the forward pad translation, and then the intrinsic 'sling' muscles that surround the base of each whisker follicle drive further protraction (*Dörfl, 1982*; *Berg and Kleinfeld, 2003*; *Hill et al., 2008*; *Bosman et al., 2011*; *Haidarliu et al., 2015*). Sensory signals arising from whisker-object contact are transmitted through the infraorbital nerve to trigeminal ganglion, ventral-posterior medial (VPM) thalamus, and S1 (*Petersen, 2007*). Because whisker pad muscles almost completely lack spindles, proprioception is thought to play a minor role, if any, in determining whisker position in space. Instead, sensations that arise from whisker self-motion – 'reafferent' signaling – is thought to play an important role in determining whisker position and object localization (*Kleinfeld and Deschênes, 2011*). Although earlier studies suggested that reafferent signaling is encoded in parallel to afferent (touch-related) signals via posterior medial (POm) thalamus (*Yu et al., 2006*), recent evidence suggests that reafferent signaling is also processed along the same lemniscal (VPM-S1) pathway to cortex as afferent input from whisker-object contact (*Moore et al., 2015*).

**eLife digest** Mice use their whiskers to sense their environment and to detect nearby objects. Rather than simply allowing their whiskers to brush passively against objects, mice move them in rhythmic bursts in a process called whisking. Whisking enables neuroscientists to study how the brain gathers and processes actively acquired sensory information. However, controlling active whisker movements in the laboratory has proven challenging.

Park et al. now offer a solution based on a technique called optogenetics. The new procedure involves introducing the gene for a light-sensitive ion channel into the facial muscles of the mouse. Shining blue light onto the area of skin where the whiskers grow – the whisker pad – activates these ion channels. Park et al. were able to use this technique to trigger the contraction of the facial muscles and the movement of the whiskers. Furthermore, stimulating different muscles in different areas of the whisker pad produced either forward or backward whisker movements. The strength of the whisker movements varied with the intensity of the light, and with how often and for how long light was applied.

Recordings of neural activity showed that sensory signals from light-induced whisker movements reach the same region of the brain as signals from natural whisker movements. Behavioral experiments showed that mice could perceive these whisker movements, despite the fact that they did not generate them.

By establishing a method for triggering whisker movements on demand, Park et al. have provided a convenient way of investigating active sensory processing. In addition, this method opens up new possibilities for using optogenetics after injury or degeneration of the nerves that control movement. Ultimately, by using light to trigger muscle contraction directly, it may be possible to restore movement in individuals who have sustained nerve damage through injury or disease.

Many studies of reafferent sensory signaling have used an 'artificial whisking' paradigm to elicit whisker movements in anesthetized rodents by electrical stimulation of the buccal branch of the facial motor nerve (*Zucker and Welker, 1969*; *Brown and Waite, 1974*; *Szwed et al., 2003*). Artificial whisking produces whisker protractions with amplitude and frequency that can be well controlled experimentally. This paradigm has drawbacks, however, including the necessity to perform experiments in anesthetized subjects, which makes it difficult to relate reafferent signaling to behavior; and the inability to stimulate certain muscle groups, which means that only whisker protractions, not retractions, can be evoked. Recently, optogenetic studies of motor nerves and muscles have used the hindlimb as a model system (*Liske et al., 2013*; *Towne et al., 2013*; *Bryson et al., 2014*; *Magown et al., 2015*). While various central elements of the whisker system have been targeted for optical control in behaving mice (*Poulet et al., 2012*; *O'Connor et al., 2013*; *Sachidhanandam et al., 2013*; *Matyas et al., 2010*), peripheral optogenetic stimulation have not been used to investigate control of whisker movements.

In this study, we report that optogenetic stimulation of the whisker pad in Emx1-Cre;Ai27D transgenic mice evokes whisker movements due to channelrhodopsin-2 (ChR2) expression in select intrinsic and extrinsic muscles. We first characterize the amplitude and frequency of whisker protractions evoked by anterior whisker pad stimulation in anesthetized mice. We then compare the electrophysiological responses in S1 to optogenetic and mechanical whisker stimulation. Finally, we show that awake, head-fixed mice are able to perceive optical whisker pad stimulation in a behavioral detection task. The results suggest that optogenetic stimulation of whisker pad muscles leads to sensory perception through reafferent signaling.

## Results

### Optically evoked whisker movements

In initial screens of adult Emx1-Cre;Ai27D mice (offspring of crossing Emx1-Cre and Ai27D lines), we discovered that whisker movements were evoked by blue light directed toward the whisker pad.

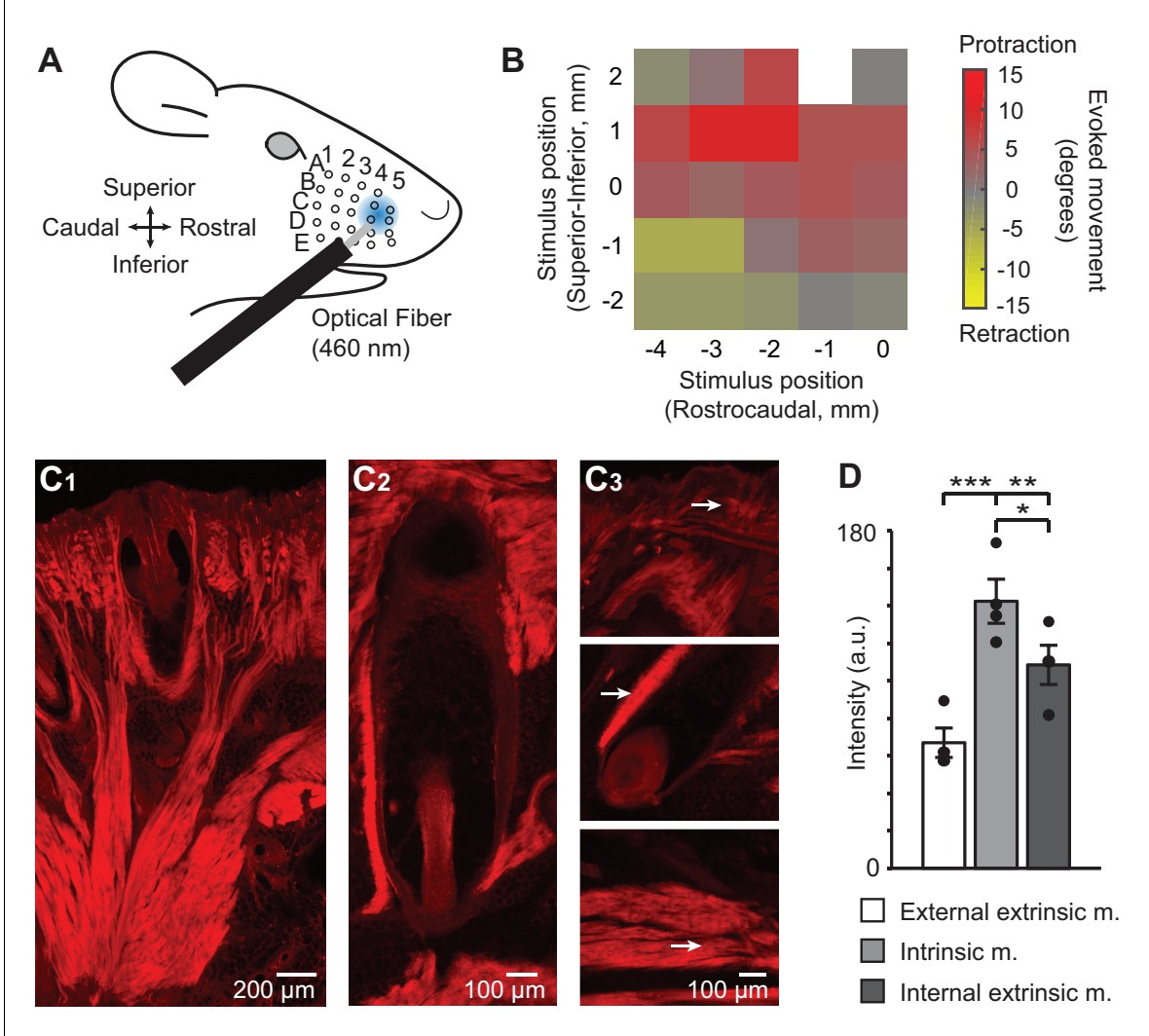

**Figure 1.** ChR2/tdTomato expression in whisker pad muscles and optical activation of whisker movements in Emx1-Cre;Ai27D mice. (**A**) Illustration of experimental setup showing the position of 460 nm light spot on the whisker pad used for optogenetic stimulation. (**B**) Example color map from one mouse (isoflurane anesthesia, 0.8–1.5%) showing the direction of movement (rostral or caudal) of whisker B2 evoked by optogenetic whisker pad stimulation at different locations on the pad. The maximum amplitude of whisker movement in degrees is color coded for each position tested. White pixel indicates location with no measurement. Whisker protractions evoked by rostral optical stimulation were the focus of the present study. (**C₁**) Intrinsic and extrinsic muscles of the whisker pad exhibit tdTomato fluorescence, as seen in histological sections. Photomicrographic montage of tdTomato fluorescence in a coronal section of the mystacial pad. Scale bar 200 µm. (**C₂**) Photomicrographic montage of tdTomato fluorescence in a whisker follicle in a transverse section. Scale bar 100 µm. (**C₃**) Representative photomicrographs of tdTomato fluorescence used for quantification (as in D). Arrows point to regions of quantification for the external extrinsic protractor muscles (top; pars maxillaris superficialis and pars maxillaris profunda of *M. nasolabialis profundus*), intrinsic follicular muscle (middle), and internal extrinsic retractor muscles (bottom; pars media superior and pars media inferior of *M. nasolabialis profundus*) following the terminology of *Haidarliu et al., 2015*. Scale bar 100 µm. (**D**) Summary of ChR2/tdTomato fluorescence intensity. Fluorescence intensity was lowest in external extrinsic muscle and highest in intrinsic muscle. Bars are mean ± SEM and individual data points are plotted. ***p<0.001, **p<0.01, *p<0.05.

The following figure supplements are available for figure 1:

**Figure supplement 1.** Analysis of retraction and protraction movements for individual whiskers.

**Figure supplement 2.** Absence of ChR2/tdTomato expression in vibrissal nerve fibers of EMX-cre;Ai27D mice.

While cortical expression of ChR2 in Emx1-Cre;Ai27D (or the similar Emx1-Cre;Ai32) mice is well known (**Madisen et al., 2012**; **Zagha et al., 2013**; **McAlinden et al., 2015**), the functional properties of incidental peripheral expression have not been characterized. Therefore, our goals were to determine 1) the localization of ChR2 expression in the whisker pad in Emx1-Cre;Ai27D mice and the functional properties of whisker movements evoked by peripheral optogenetic stimulation; 2) whether peripheral optogenetic stimulation activates S1 in a fashion similar to mechanical whisker stimulation; and 3) whether peripheral optogenetic stimulation induces behaviorally reported sensory detection.

We first characterized the whisker movements evoked by a 2–3 mm diameter, 460 nm spot of light aimed at different regions of the whisker pad in anesthetized Emx1-Cre;Ai27D mice (isoflurane 0.8–1.5%) (**Figure 1A**). The direction of movement depended on the location of the spot, such that illumination of the rostral pad resulted in whisker protraction, while illumination of the caudal-inferior pad resulted in whisker retraction (**Figure 1B**). Stimulation at some locations elicited more complex combinations of protractions and retractions from individual whiskers (**Figure 1—figure supplement 1**; see also **Video 1**, **Video 2**, **Video 3**). These regional variations in light-evoked protraction and

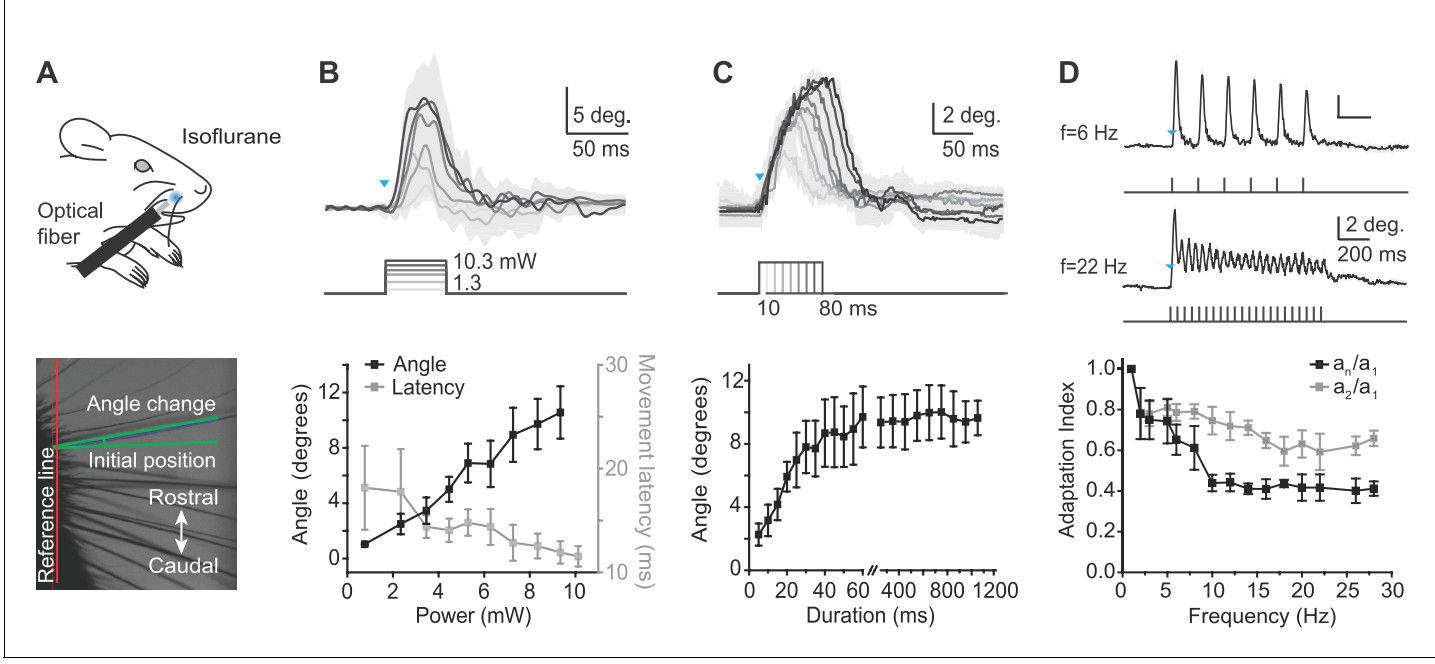

**Figure 2.** Characterization of whisker movements evoked by optogenetic whisker pad stimulation. (**A**) Top: Illustration of experiment setup (isoflurane anesthesia, 0.8–1.5%). Rostral whisker pad illumination (460 nm) was used to evoke whisker protractions (positive angle values). Bottom: Image of whiskers under infrared illumination as used for whisker tracking. Angle changes of individual whiskers were measured relative to the initial position (green lines). (**B**) Relationship between light intensity and evoked whisker protractions. Top: Example traces from one mouse (mean ± SD of single trial for n = 4 whiskers). Blue triangle indicates the onset of the light stimulation. Intensities: 1.3, 3.1, 4.1, 6.5, 8.4, 10.3 mW. Duration of stimuli, 50 ms. Bottom: amplitude of evoked angle change (left axis) and movement latency (right axis) vs. light intensity (bin size, 1 mW; mean ± SEM; n = 4 mice). (**C**) Relationship between light duration and evoked whisker protractions. Top: Example traces from one mouse (mean ± SD of 10 trials). Blue triangles indicates the onset of the light stimulation. Durations: 10–80 ms at 9.94 mW intensity. Bottom: amplitude of evoked angle change vs. light duration (mean ± SEM; n = 4 mice; note gap in axis between 60 and 250 ms and difference in x-axis scaling for 5–60 ms and 250–1200 ms). (**D**) Adaptation of evoked whisker protractions to optical pulse frequency. Top: two example traces from one mouse at 6 Hz and 22 Hz stimulation (9.94 mW). Blue triangle indicates the onset of the light stimulation. Bottom: Adaptation indexes (black: $a_n/a_1$, ratio of last to first response amplitude; gray, $a_2/a_1$, ratio of second to first response amplitude) plotted versus stimulus frequency (mean ± SEM; n=3 mice).

The following source data is available for figure 2:

**Source data 1.** Data for **Figure 2B**.
**Source data 2.** Data for **Figure 2C**.
**Source data 3.** Data for **Figure 2D**.

retraction were qualitatively similar in 5 of 5 mice tested. In the rest of this study, we focused on whisker protractions evoked by optogenetic activation of the rostral whisker pad.

We performed histological analysis to determine if these results could be explained by expression of ChR2 in muscles that control different types of whisker movements (*Dörfl, 1982*; *Hill et al., 2008*; *Haidarliu et al., 2015*). Indeed, analysis of the native fluorescence of the ChR2/tdTomato fusion protein in sections of the whisker pad revealed tdTomato expression in intrinsic and extrinsic whisker pad muscles (*Figure 1C*). Intrinsic muscles appeared on both superior and inferior sides of the follicle in coronal sections, and on the rostral side in transverse sections (*Figure 1C$_1$, C$_2$*), consistent with their sling-like morphology (*Dörfl, 1982*; *Haidarliu et al., 2010*). No fluorescence was evident in the infraorbital (sensory) nerve (*Figure 1—figure supplement 2*). Comparison of fluorescence intensity indicated the highest intensity in intrinsic sling muscles, followed by the deep extrinsic retractor muscle (pars maxillaris superficialis and pars maxillaris profunda of *M. nasolabialis profundus*), and lastly the superficial extrinsic protractor muscle (the pars media superior and pars media inferior of *M. nasolabialis profundus*,) ($F_{(2,6)}$ = 57.66, p=0.0001, repeated measures ANOVA followed by paired contrasts; p=0.0004 comparing external extrinsic and intrinsic, p=0.0086 comparing external extrinsic and internal extrinsic, p=0.0341 comparing intrinsic and internal extrinsic; n=4 follicles from one mouse) (*Figure 1D*). These results indicate that light-evoked whisker movements in Emx1-Cre;Ai27D mice arise from activation of ChR2 expressed in extrinsic and intrinsic whisker pad muscles.

To quantitatively characterize the whisker movements evoked by peripheral optogenetic stimulation, we recorded high-speed video (500 frames/s) in anesthetized mice (isoflurane 0.8–1.5%) (*Figure 2A*) in response to 460 nm light stimulation of varying intensity, duration, and frequency, with illumination centered at the rostral protraction area. The amplitude of whisker protraction in response to a 50 ms light pulse of increasing intensity (range, 1.3–10.3 mW) increased approximately linearly to a maximum amplitude of 11.4 ± 1.2 degrees (mean ± SEM, n = 4 mice; maximum 14.8 degrees in one mouse; *Figure 2B*). In 2 of 4 mice, the angle change appeared to saturate at less than maximal power (8.37 and 9.94 mW, respectively). The average latency of optogenetically evoked whisker movement was 13.5 ± 0.3 ms (mean ± SEM, n = 4 mice; threshold defined as 10% of the maximum peak) and was not affected by stimulus duration. We used 9.94 mW to define the relationships between whisker protraction, duration, and frequency (below). We next measured the relationship between whisker protraction and optical stimuli of varying duration from 5 to 1200 ms (at 9.94 mW intensity). Whisker protraction angle increased with the duration of the optical stimulus, saturating with durations longer than approximately 60 ms (*Figure 2C*).

Finally, we tested whether whisker protractions could follow 1 s long trains of optogenetic stimulation of varying frequencies, from 1 to 45 Hz (*Figure 2D*), covering the frequency range of natural exploratory whisking (*Welker, 1964*; *Carvell and Simons, 1990*; *Harvey et al., 2001*). The duration of each pulse in the train was 2 ms. Two alternative adaptation indexes were calculated as either the ratio of amplitudes of the last response to the first response in the pulse train ($a_n/a_1$) or the amplitudes of the second response to the first response in the pulse train ($a_2/a_1$). A smaller adaptation index indicates a larger difference between first and second or last peaks during the optical pulse train, and therefore greater adaptation. The adaptation index of $a_n/a_1$ decreased to lower values than the adaptation index of $a_2/a_1$ at the same frequency (e.g., at f = 28 Hz, $a_n/a_1$ = 0.41 ± 0.04, $a_2/a_1$ = 0.66 ± 0.04), indicating further adaptation with increasing number of pulses (*Figure 2D*). At frequencies greater than 30 Hz, individual evoked movements were no longer discernible, although the envelope of the angle change continued to show adaptation up to 45 Hz and became similar to movements elicited by constant prolonged light steps (data not shown). These data indicate that optogenetically evoked whisker protractions show activation and adaptation over a behaviorally relevant range of frequencies. Together, the results of *Figure 2* define fundamental stimulus parameters for optogenetic activation of whisker protractions in Emx1-Cre;Ai27D mice.

## Peripheral optogenetic stimulation evokes activity in S1

We next investigated whether peripheral optogenetic stimulation evoked neural activity in S1 by implanting 8-channel microwire arrays in S1 of Emx1-Cre;Ai27D mice. After one week of recovery, we recorded local field potentials (LFPs) and multiunit spiking activity in three anesthetized mice (isoflurane 0.8–1.5%) in response to peripheral optogenetic stimulation (*Figure 3A*). To account for potential differences in the locations of the arrays in S1, we analyzed signals from the channel with

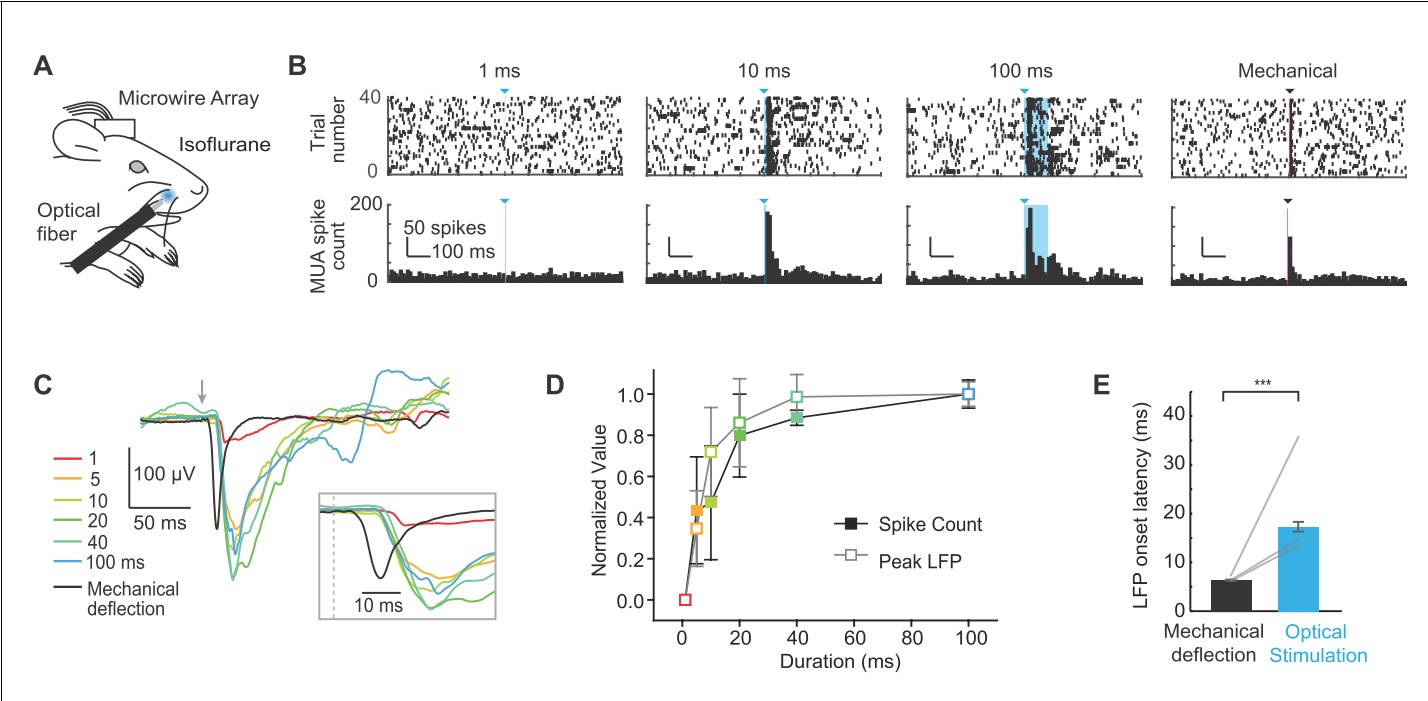

**Figure 3.** Extracellular recordings of S1 activity in response to optogenetic whisker pad stimulation. (**A**) Illustration of experiment setup (isoflurane anesthesia, 0.8–1.5%), including chronically implanted microwire array. (**B**) Example peri-stimulus time histograms (PSTHs; bin size, 10 ms) for one mouse displayed ± 0.5 s relative to stimulation onset. Blue triangles and lines denote onset of 460 nm light stimulation; black triangle and line denotes onset of mechanical whisker stimulation. (**C**) Example local field potentials (LFPs) from one channel in response to optical whisker pad stimulation of various durations (1–100 ms) and mechanical stimulation of whisker C3. Each trace is the mean of 30 trials. (**D**) Peak LFP and maximum spike count (mean ± SEM, n = 3 mice), normalized to the maximum response for each channel. The channel that showed the largest response was selected from each mouse. (**E**) Comparison of LFP response latency for peripheral optical stimulation and mechanical whisker stimulation (shortest latency channel selected for each mouse). Bar graphs show mean (n = 3 mice) and lines connect individual subjects. Mean latency was 17.3 ± 1.0 ms (mean ± SEM) for 20–100 ms optical stimuli and 6.5 ± 0.1 ms (mean ± SEM) for mechanical whisker stimulation. ***p<0.001.

The following source data is available for figure 3:

**Source data 1.** Data for *Figure 3D*.

the shortest latency in each mouse. Spiking activity and LFP amplitude increased with the duration of peripheral optogenetic stimulation (*Figure 3B,C*). Plotting the responses normalized to the maximum response in each mouse indicated that spike count and LFP amplitude increase steeply with light pulse duration from 1–20 ms, and moderately between 20–100 ms (*Figure 3D*). We used a brief mechanical deflection of the whisker to compare S1 response timing. While the active whisker protraction evoked by optogenetic stimulation provides qualitatively distinct activation of sensory input compared with passive mechanical deflection, this experiment allowed us to determine the relative latencies of S1 responses. The spike number and LFP amplitude evoked by peripheral optogenetic stimulation were on average similar to those evoked by mechanical whisker deflection (peak spike number per stimulus in 10 ms bin: 3.5 ± 0.2 mechanical, 3.2 ± 1.1 optical; LFP peak amplitude: −168.5 ± 24.7 μV mechanical, 159.66 ± 45.1 μV optical). In one mouse, the largest responses observed to peripheral optogenetic stimulation were 5.8 spikes/stimulus and −267.5 μV peak LFP amplitude (129.3 ± 16.3% of the LFP amplitude evoked by mechanical whisker stimulation; values were 54.0 ± 7.6%, and 68.4 ± 2.6% in two other mice). Notably, the response latency determined from LFP recordings was 10.8 ± 0.1 ms longer for peripheral optogenetic stimulation compared to mechanical whisker stimulation ($p<1\times10^{-5}$ in n = 3/3 mice; paired t-tests; 13–39 trials per mouse) (*Figure 3C*, inset; *Figure 3E*). These results suggest that the longer latency in S1 for peripheral optogenetic stimulation can likely be attributed to the 11.9 ± 0.8 ms delay associated with the initiation of evoked whisker movement (*Figure 2B*; value from 9.3 mW intensity), and that sensory signals

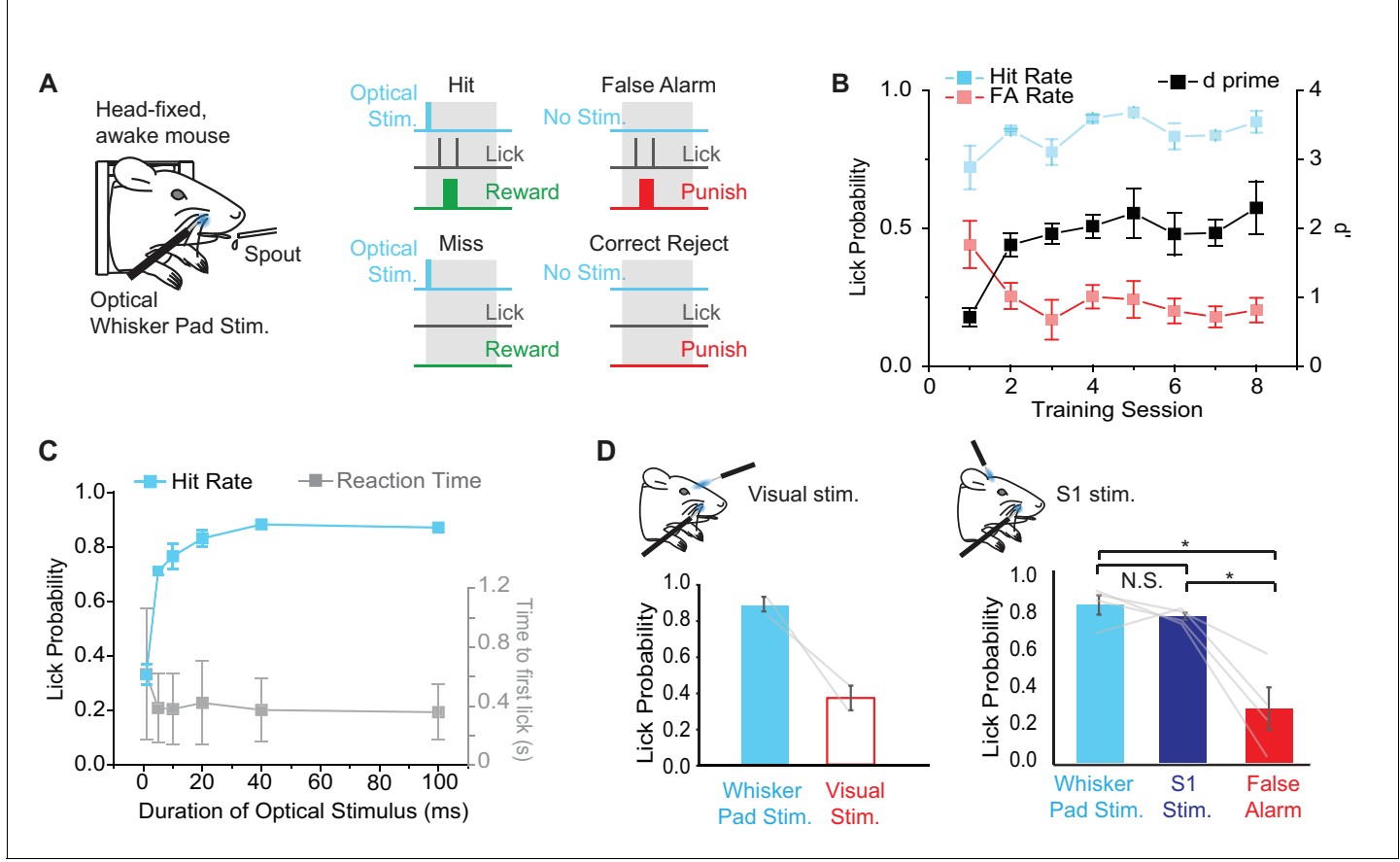

**Figure 4.** Behavioral performance in mice trained to detect optogenetic stimulation of the whisker pad. (**A**) Illustration of the behavioral task. Water deprived, head-fixed mice were rewarded with water for licking within a 2 s response window (gray boxes) after optical stimulation (460 nm) of the rostral whisker pad (Hit trials). Licking in the absence of stimulation resulted in a False Alarm (FA) and punishment (tone and/or 2M salt water). (**B**) Changes in behavioral performance with training. A maintained Hit rate (blue) and reduced FA rate (red) accounted for the increase in performance (d′; black) over sessions (mean ± SEM, n = 4 mice). Note that learning curves for 2 of 4 mice are shown from the time of introduction of salt water punishment for FA and are aligned to first learning session (d′>1). (**C**) Dependence of Hit rate (left axis) and reaction time (right axis) on stimulation duration. In expert mice, optical whisker pad stimuli of various durations were included randomly on 10% of trials during behavioral sessions (mean ± SEM; n = 4 mice). (**D**) A second optical fiber delivered either visual stimulation (left) or S1 optogenetic stimulation (right) on a random 10% of trials (100 ms duration for all stimuli). Lick probability was reduced for visual stimuli (left), but not S1 stimulation (right). *p<0.05; N.S., not significant.

The following source data is available for figure 4:

**Source data 1.** Data for *Figure 4B*.

**Source data 2.** Data for *Figure 4C*.

arrive rapidly in S1 once whisker movement is initiated. These results are consistent with our histological data suggesting that ChR2 is expressed in muscle and absent from sensory nerve (*Figure 1C*; *Figure 1—figure supplement 2*).

## Behavioral report of peripheral optogenetic stimulation

In order to determine whether Emx1-Cre;Ai27D mice can perceive peripheral optogenetic stimulation, we established a modified head-fixed sensory detection task inspired by recent studies (*O'Connor et al., 2013*; *Sachidhanandam et al., 2013*) (*Figure 4A*). We trained mice to report the presence of 100 ms (9.94 mW) peripheral optogenetic stimulation by licking for water reward. In each Hit trial, mice received a water drop for licking within a 2 s time window after a stimulus. False

Alarm (FA) trials occurred if the mouse licked when no light was delivered during the stimulus time window and resulted in presentation of a 2 s, 5 kHz tone and 5–10 s time out before the next trial. Inter-trial time randomly varied from 5–10 s. Two of four mice learned the task (d'>1) within 4 sessions (2 sessions per day, 125 trials per session), showing maintained Hit rate with relatively low FA rate (FA rate < 0.3). The other two mice learned only after introducing 2 M salt water solution as additional punishment for licking during FA trials. We aligned learning curves for all mice relative to the start of learning (d'>1) (*Figure 4B*), which was after the introduction of salt water punishment in 2/4 mice. Overall, behavioral performance improved via a maintained high Hit rate and a decrease in FA rate, resulting in an increase in d' from 0.7 ± 0.1 to 2.3 ± 0.4 (mean ± SEM; p = 0.017, paired t-test; n = 4 mice) over the course of training (*Figure 4B*).

In order to determine the psychometric curve for peripheral optogenetic stimulation, we varied the duration of optical stimulation (stimuli from 1–100 ms presented randomly with equal probability) in expert mice (d'>1.5) and tested the effects on task performance. We found that Hit rate fell to chance levels with stimuli shorter than 5 ms (*Figure 4C*), defining a lower limit of optical stimulation necessary for behavioral detection. Note that this behaviorally measured detection threshold is similar to the threshold for evoked whisker movements (*Figure 2C*) and S1 activity (*Figure 3D*) measured in anesthetized mice.

Although the fiber tip was shielded, we performed additional controls to rule out the possibility that mice were responding to visual stimulation arising from the optogenetic excitation light. In expert mice (d' > 1.5), we used a second optical fiber placed in front of the mouse's head to deliver diffuse blue light to the eye on the same side of the face. Visual catch trials were added to the training regime with 10% probability (probability of whisker pad stimulation remained at 50%). Lick rate for visual catch trials was 0.37 ± 0.07 compared to 0.87 ± 0.04 for peripheral optogenetic stimulation (328 peripheral stimulation trials, 75 visual catch trials in 6 sessions from n = 2 mice) (*Figure 4D*), suggesting that visual stimulation was not a salient cue involved in performance of the peripheral optogenetic detection task.

We next tested whether S1 neural activity, which is elicited by peripheral optogenetic stimulation (*Figure 3*), is sufficient for task performance in mice trained to detect peripheral stimulation. We delivered 460 nm optogenetic stimulation to S1 through a cranial window (implanted in the initial surgery; see Materials and methods) in 10% of trials using a second optical fiber. Additional introduction of ChR2 was unnecessary because Emx1-Cre;Ai27D mice express ChR2 in cortical pyramidal neurons. In expert mice, lick rate in response to S1 stimulation (100 ms) was not significantly different compared to peripheral optogenetic stimulation (100 ms), but was significantly greater than FA lick rate ($F_{(2,6)}$ = 17.62, p=0.0031, repeated measures ANOVA followed by paired contrasts; p=0.43072 comparing peripheral and S1 stimulation, p=0.0238 comparing S1 stimulation and FA, p=0.016 comparing peripheral stimulation and FA; 160 peripheral stimulation trials, 156 S1 simulation trials, 160 FA trials in 2 sessions from n = 4 mice) (*Figure 4D*). These results indicate that optogenetic stimulation of S1 is sufficient to drive sensory detection in mice that were trained to detect peripheral optogenetic stimulation.

## Discussion

We found that optogenetic stimulation of mystacial pad muscles in Emx1-Cre;Ai27D mice induces whisker movements linked to ChR2 expression in intrinsic and extrinsic muscles, and that mice can readily report sensory perception associated with peripheral optogenetic stimulation. Combined cortical stimulation experiments illustrated the utility of Emx1-Cre;Ai27D mice for optogenetic investigation of both peripheral and central excitable cells.

### Peripheral expression in Cre driver lines

We initially discovered that Emx1-Cre;Ai27D mice express ChR2/tdTomato in peripheral tissue by examining pups using fluorescence goggles. A practical benefit of peripheral expression worth mentioning is that transgene transmission to offspring can be inferred simply by visual inspection under fluorescence instead of traditional DNA genotyping. Peripheral Cre expression is known in other Cre driver lines commonly used for neurobiological studies of the central nervous system. For example, Chat-Cre, often used to target cholinergic neurons of basal forebrain (*Eggermann et al., 2014*; *Hangya et al., 2015*) also drives expression in motoneurons (*Gong et al., 2007*; *Takatoh et al.,*

2013); PV-Cre, often used to target a class of central GABAergic interneurons (*Cardin et al., 2009*; *Sohal et al., 2009*; *Gentet et al., 2010*), also expresses in proprioceptive neurons of the dorsal root ganglion (*Hippenmeyer et al., 2005*), sensory neurons of trigeminal ganglion (*Sakurai et al., 2013*), and fast-twitch skeletal muscle fibers (*Chakkalakal et al., 2012*). It should be noted that, in addition to such incidental central/peripheral expression, many other Cre driver lines have been developed exclusively for investigation of the peripheral nervous system (*da Silva et al., 2011*; *Rutlin et al., 2014*) and muscle (*Chen et al., 2005*; *Li et al., 2005*; *Chakkalakal et al., 2012*; *Magown et al., 2015*). In our experiments, we took advantage of the central and peripheral Cre transgenic expression in Emx1-Cre;Ai27D mice to achieve optogenetic activation of facial muscles and central neurons in the same subjects (*Figure 4*).

## Optogenetically evoked whisker movements

Our study builds upon classic studies that used electrical stimulation of the buccal motor branch of the facial nerve to induce artificial whisking (*Zucker and Welker, 1969*; *Brown and Waite, 1974*; *Szwed et al., 2003*). We summarize here some of the key features of optically induced whisker movements compared to those induced by artificial whisking.

### Muscle versus nerve stimulation

Whisker movements in Emx1-Cre;Ai27D mice are induced by direct stimulation of the ChR2-expressing muscle instead of the innervating facial nerve. Although untested to our knowledge, it might be possible to activate whisker movements in existing strains of transgenic mice via optical stimulation of the facial nerve. For example, certain lines of Chat-ChR2 BAC transgenic mice are reported to express ChR2 in the facial nucleus (*Zhao et al., 2011*). Thy1-ChR2 mice have been used for sciatic nerve stimulation (*Llewellyn et al., 2010*), but it is unclear whether these mice express ChR2 in the facial nerve. However, direct muscle stimulation could be an advantage for therapeutic models in which denervation or motoneuron degeneration has occurred (*Magown et al., 2015*).

### Interpretation of muscle activation

ChR2 expression in muscle also allows an extended repertoire of whisker movements compared to artificial whisking, including both optically evoked retractions and protractions. While the current study focused on the evoked protraction, further study is warranted to quantitatively characterize evoked retractions, including the possibility of using sequential protraction-retraction optical stimulation to mimic the natural whisk cycle (*Hill et al., 2008*). The different types of movements are due to ChR2 expression in at least two specific whisker pad muscles. We identified native ChR2/tdTomato expression in intrinsic follicular muscles and in the extrinsic muscle *M. nasolabialis profundus* (*Figure 1*). Our results, in accordance with previous anatomical and physiological studies (*Dörfl, 1982*; *Dörfl, 1985*; *Berg and Kleinfeld, 2003*; *Hill et al., 2008*; *Haidarliu et al., 2010*; *2015*), suggest that light-evoked protractions involve activation of intrinsic muscles, as well as the pars media superior and pars media inferior of the *M. nasolabialis profundus*. Because ChR2 is expressed in both muscle types, the relative contribution of activation of intrinsic versus extrinsic muscles to evoked whisker protractions remains unclear. Retractions involve activation of the (deep) pars maxillaris superficialis and pars maxillaris profunda of the *M. nasolabialis profundus*. Optical stimulation at the rostral whisker pad could favor protraction because of the morphology of the muscles along the surface of the pad, specifically the caudal-to-rostral tapered morphology of the deep retraction muscles (pars maxillaris superficialis and pars maxillaris profunda) (*Haidarliu et al., 2010*). Achieving selective expression of ChR2 in extrinsic and intrinsic muscles would help to enable functional dissection of the whisker movements controlled by these muscle types.

### Amplitude and frequency of evoked protractions

The maximum amplitude of optogenetically evoked protractions that we found was 11.4 degrees on average in four mice (largest individual mouse average, 14.8 degrees, *Figure 2*; see also *Figure 1—figure supplement 1* for single trial examples from multiple tracked whiskers), while studies using artificial whisking in rats report amplitudes of up to 20 degrees (*Yu et al., 2006*; *Castro-Alamancos and Bezdudnaya, 2015*). One possible explanation is that the excitation light was restricted to a 2–3 mm diameter spot on the whisker pad, while nerve stimulation evokes

widespread muscle activation via acetylcholine release throughout the whisker pad. We also note that optogenetically evoked protractions showed stronger frequency adaptation than reported for artificial whisking. We found strong adaptation over a stimulus frequency range of 2 to 28 Hz (*Figure 2D*), whereas 100 Hz electrical nerve stimulation (artificial whisking) results in sustained whisker protraction for up to 1 s (*Castro-Alamancos and Bezdudnaya, 2015*). This could be explained by potential differences in muscle groups recruited by optogenetic stimulation in Emx1-Cre;Ai27D mice compared to artificial whisking. It is unlikely that desensitization of ChR2-mediated currents accounts for the adaptation effects we measured, since optogenetic stimulation of hindlimb muscles produces non-adapting contractions with pulse durations up to 1 s (*Magown et al., 2015*). Intrinsic muscles of the whisker pad are rapidly fatiguing because they consist almost exclusively of type 2B muscle fibers (*Jin et al., 2004*). The fiber composition of extrinsic muscles is less clear, but appears to be of mixed fiber type (*Jin et al., 2004*; *Grant et al., 2014*). Our results showing strong adaptation are consistent with activation of rapidly fatiguing muscle fibers by optogenetic stimulation. It is possible that artificial whisking recruits less fatigable extrinsic protractor muscles more strongly than optogenetic stimulation. It might be possible in future experiments to determine the relative contribution of different whisker pad muscle groups to the adaptation effects we observed using selective optogenetic stimulation of intrinsic and extrinsic muscles.

## Detection of evoked whisker movements: involvement of reafferent sensory signaling

In addition to the issues discussed above, the major benefits of peripheral optogenetic stimulation are the non-invasive activation of whisker movements using light and the ability to perform experiments without the use of anesthesia. We used these features to design a behavioral task in head-fixed mice in order to investigate whether mice can perceive whisker movements that result from peripheral optogenetic stimulation. Several recent studies have used similar behavioral paradigms to investigate afferent sensory perception using tasks designed to assess, for example, stimulus detection, object localization, texture or frequency discrimination (*Arabzadeh et al., 2005*; *O'Connor et al., 2010*; *Morita et al., 2011*; *Sachidhanandam et al., 2013*; *Musall et al., 2014*; *Chen et al., 2015*). All of these tasks were designed to test the detection of *afferent* input, that is, aspects of sensory input arising from an external stimulus. The goal of our task was to test the detection of *reafferent* input, that is, sensory input arising from self-generated movement. During natural whisking, reafferent input arises from the whiskers moving through space and, because rodents mostly lack proprioceptors in whisker pad muscles (*Moore et al., 2015*), reafferent signaling is considered important for encoding whisker position and locating objects in space (*Kleinfeld and Deschênes, 2011*). However, reafferent signaling has been difficult to study: either the subjects are anesthetized and well controlled whisker movements are elicited by artificial whisking, or the subjects are awake are freely whisking, where whisker movements are not under experimental control. Thus, our behavior task provided a unique opportunity to investigate the detection of reafferent signaling in awake animals with well controlled stimuli. We found that mice could readily learn to detect peripheral optogenetic stimulation (*Figure 4*). It should be noted that two mice required introduction of salt water punishment to reduce impulsive responding, as has been used in other types of Go/NoGo tasks (*Rebello et al., 2014*), and that d' remained below that observed in other studies due to a sustained FA rate of approximately 0.2 (*Huber et al., 2012*; *Chen et al., 2015*). The reason for the sustained FA rate is not clear but could relate to motivation or hydration levels (*Guo et al., 2014*) that could be further optimized in future studies. Similar to afferent sensory detection tasks, behavioral performance improved via maintained Hit rate and reduced FA rate over days. Four lines of evidence suggest that mice indeed used reafferent sensory input to perform the behavioral task. (1) Electrophysiological recordings from S1 showed that the latency was approximately 10 ms longer for optogenetically compared to mechanically evoked responses (*Figure 3C,E*), suggesting that the whisker must move before sensory signals arrive in cortex. (2) Changes in evoked whisker movements (*Figure 2C*), neural signals in S1 (*Figure 3D*), and behavioral responses (*Figure 4C*) showed similar relationship with the duration of optogenetic whisker pad stimulation. (3) Visual stimulation was not sufficient to substitute for peripheral stimulation in expert mice performing the detection task, suggesting that mice were not responding to visual aspects of optogenetic stimulation. (4) Optogenetic stimulation of S1 was sufficient to substitute for peripheral optogenetic stimulation in the detection task (whereas naïve mice did not

respond to S1 stimulation; data not shown), suggesting that S1 is involved in perception of reafferent sensory signals.

## Conclusions and future applications

We conclude that the whisker movements elicited by optogenetic activation of muscles in the whisker pad lead to sensory perception through reafferent sensory signaling. Optogenetic whisker pad stimulation provides new opportunities for studies of sensorimotor integration in behaving mice. In the future, the non-invasive nature of peripheral optogenetic stimulation could be used to further investigate reafference and whisker-object contact during evoked protractions and retractions. Furthermore, the ability to stimulate muscle directly could have therapeutic benefits in preclinical studies of motor recovery after peripheral nerve injury or motoneuron degenerative disorders.

# Materials and methods

## Subjects

Homozygous Emx1-Cre mice (Stock no. 005628, Jackson Laboratory, Bar Harbor, ME) were crossed with homozygous Ai27D mice (Stock no. 012567, Jackson Laboratory), and the resulting Emx1-Cre; Ai27D offspring (heterozygous for both transgenes) were used for experiments. PV-Cre;Ai27D mice (Stock no. 008069, Jackson Laboratory) in *Figure 1—figure supplement 2* were generated similarly. Ai27D mice express a ChR2(H134R)/tdTomato fusion protein in a Cre-dependent manner (*Madisen et al., 2012*). Emx1-Cre mice, which express Cre recombinase from the Emx1 locus (*Gorski et al., 2002*), have been found to express Cre in limited peripheral tissues (http://www.informatics.jax.org/) in addition to the better known Cre expression in forebrain glutamatergic neurons (*Madisen et al., 2012*; *Zagha et al., 2013*; *McAlinden et al., 2015*).

## Surgical preparation

All procedures were approved by Rutgers University Institutional Animal Care and Use Committee (IACUC; protocol 13–033). Male Emx1-Cre;Ai27D mice were implanted with a glass cranial window and metal head post, as in previous work (*Holtmaat et al., 2009*; *Margolis et al., 2012*). Briefly, 4–9 week old mice were anesthetized with isoflurane (4% induction, 0.8–1.5% maintenance) and placed on a feedback controlled heating blanket maintained at 36°C (FHC, Bowdoin, ME) mounted on a stereotaxic frame (Stoelting, Wood Dale, IL). After cleaning the surface of the skull, bonding agent (iBond, Heraeus Kulzer, Hanau, Germany) and a thin layer of dental cement (Tetric Evoflow, Ivoclar Vivadent, Schaan, Lichtenstein) were applied covering the right side skull and the anterior and posterior left side skull. A 4 mm craniotomy was made with a dental drill (Osada EXL-M40, Los Angeles, CA), leaving the dura mater intact, centered approximately over S1 barrel cortex (-1 mm posterior, -3 mm lateral from Bregma). A 4 mm diameter #1 thickness circular cover glass (Menzel Glaser, Braunschweig, Germany) was implanted directly on the dura. The edges of the glass window were covered with dental cement, and the junction between glass and cement was sealed with cyanoacrylate glue. A custom metal head post was cemented to the right side skull. After surgery, mice were housed under a reversed light cycle (lights off 08:00–20:00) and had free access to food and water. All subsequent experiments were conducted during the dark phase of the light cycle. Beginning 3–4 days after surgery, mice were handled daily by the experimenter. Adaptation to head restraint began after at least one week of recovery. For behavioral training (below), mice were water restricted to 1 ml/day.

## Peripheral optogenetic stimulation and whisker tracking in anesthetized mice

Five mice underwent peripheral optogenetic stimulation under isoflurane anesthesia (4% induction, 0.8–1.5% maintenance). In each session, whisker movements were recorded in response to one stimulus parameter (intensity, duration, frequency; three total sessions per mouse). One mouse was excluded from the study because of anesthesia-related whisker motion artifacts in the first session; another mouse was excluded from the varying frequency experiment because of changing baseline whisker position. Mice were stabilized by bolting the head post to a cross bar and placed on a feedback controlled heating pad maintained at 36°C. Lack of reflex to tail/foot pinch was used to assess

adequate anesthesia levels. Isoflurane levels were adjusted during recordings to maintain an approximately 1 Hz respiration rate. This was a useful benchmark for adequate but light anesthesia and helped to avoid respiration-associated whisker movements that interfered with measurements of evoked whisker movements. Optogenetic stimulation was provided by a high-powered 460 nm LED (Prizmatix, Givat-Shmuel, Israel) coupled to a multi-mode optical fiber (200 µm core, 0.22 NA; Thorlabs, Newton, NJ). The duration and frequency of light pulses were controlled by trains of TTL pulses from an Arduino Uno board to the LED current driver. The fiber tip was mounted on a micromanipulator (Narishige, Tokyo, Japan) and placed within 2 mm of the anterior right side the whisker pad, resulting in a 2–3 mm diameter spot covering 3–5 whiskers. Maximum power was 10.3 mW measured at the fiber tip with a power meter (Thorlabs).

For whisker tracking, the right side whiskers were illuminated from below with an infrared LED array (850 nm, Advanced Illuminations, Rochester, VT; Edmund Optics #66–802). Evoked whisker movements were imaged through a telecentric lens (0.36x, f/6-f/18; Edmund Optics #58–257, Barrington, NJ) with a CMOS camera at 500 Hz frame rate (DR1, Photofocus AG, Lachen, Switzerland) and acquired using Streampix software (NorPix, Montreal, Canada). Each movie was 5 s in duration, including a 1 s pre-stimulus baseline. 10 trials were recorded for each stimulus parameter with an 8 s inter-trial interval. Movie files were converted to AVI format offline, and MATLAB-based (Mathworks, Natick, MA) whisker tracking software (*Knutsen et al., 2005*) was used to extract the frame-wise whisker angle for each trial. Further analysis was carried out using custom routines in MATLAB. Well-isolated individual whiskers were manually selected for whisker tracking, and in some cases traces were averaged from 3–4 tracked whiskers. Stimulus-response curves were determined by measuring changes in whisker angle relative to a manually determined resting position. In intensity plots, data were binned in 1 mW bins to account for slight intensity differences used in different subjects. Data are shown as mean ± SEM for the four mice included in the analysis. Results from one of the four mice use for intensity and duration measurements was excluded from the analysis of adaptation because of unstable data.

## Cortical electrophysiological recordings

Custom microwire arrays were implanted in S1 (4 x 2 array of 50 µm diameter, 1 mm length stainless steel microwires; 500 µm between-channel, 300 µm between-row spacing; Micro Probes, Inc., Gaithersburg, MD). Implants were performed in four of the same mice used for whisker tracking and behavior experiments (below); one of four mice was excluded due to poor signal quality. The glass window was removed (*Goldey et al., 2014*) by carefully drilling the edge of the dental cement and lifting the cover glass with blunt forceps. To allow access for the array, the dura mater was punctured with a 34 gauge needle tip. The most posterior electrodes were targeted to C2-C3/D2-D3 barrel columns, as mapped by intrinsic optical signal imaging through the cranial window before removal. The microwires were inserted by stereotaxic manipulator to 500–600 µm depth. The reference electrode was located at the anterior part of the array, 3–4 mm from the most posterior part of the array. The ground wire was inserted near the olfactory bulb through a small craniotomy and fixed with a stainless steel microscrew. The array and ground wire were stabilized with dental cement, leaving the Omnetics connector exposed. Electrophysiological measurements were made with a 32 channel amplifier (ME32, Multi Channel Systems, Reutlingen, Germany) sampled at 25 kHz. Raw data was analyzed with custom routines in MATLAB. Local field potential (LFP) data was bandpass filtered from 0.1 to 300 Hz, and spiking data from 300 Hz to 10 kHz. Multiunit spikes were detected using a threshold of 3.5 * SD of the entire recorded voltage per trial. Mechanical whisker stimulation was delivered by inserting a single whisker into a 23 gauge metal tube that was glued to a piezoelectric bending element (Physik Instrumente PL140, Karlsruhe, Germany). A 1 ms TTL pulse to the piezo current driver (Physik Instrumente E650) triggered a brief rostrocaudal whisker deflection. 30 trials were averaged for each stimulus in stimulus-response curves.

## Behavioral detection task

Optogenetic stimulus delivery was provided by the same optical fiber as in whisker tracking experiments (above). The tip of the fiber was shielded with blackout tape and placed approximately 2 mm from the right side whisker pad without touching the skin or whiskers. To preclude visual detection of the 460 nm light, blackout tape was attached to a probe and installed in front of the right eye,

and ambient green light (530 nm) was used to flood the behavior setup. Custom Arduino routines controlled the timing and structure of trials, including triggering of the LED current driver, detection of lick timing from a capacitive touch sensor coupled to the lick spout, and triggering of the water valve.

Water deprived Emx1-Ai27D mice were trained to lick for water reward in response to 100 ms peripheral optogenetic stimulation delivered to the rostral whisker pad. There was no cue for trial initiation; inter-trial interval was randomized from 5 to 10 s. 'Go' trials, when a stimulus was present, could result in either a 'Hit' or a 'Miss' behavioral response. If licks occurred within a 2 s response window after optogenetic stimulation, the trial was recorded as a Hit and mice received a 5 µl water reward. Miss trials occurred when the mouse failed to lick within the post-stimulus response window. 'NoGo' trials, when no stimulus was delivered, could result in either a 'False Alarm' (FA) or 'Correct Rejection' (CR) behavioral response. A FA response occurred when the mouse licked during the response window of a NoGo trial. Mice were punished for FAs by 2 s presentation of a 5 kHz tone followed by 5–10 s timeout (in addition to the inter-trial interval) before the next trial. A CR was recorded when mice did not lick when no stimulus was delivered. Mice were trained using 125 trials per session, two sessions per day. The probability of Go trials was lowered from 60% to 50% upon reaching a FA rate of <0.3. Mice with high FA rate after one week of training received salt water (2M NaCl) dispensed from a second spout as additional punishment (*Rebello et al., 2014*).

## Psychometric curves, catch trials, and cortical stimulation

We used d' (d prime) to measure behavioral performance, defined as z-score (Hit rate) – z-score (FA rate), where Hit rate and FA rate are probabilities and z-scores are calculated from a standard normal distribution (mean = 0) with unit variance. To calculate learning curves, the group mean changes in d', Hit rate, and FA rate were aligned in each mouse to the first session where d' remained >1 in subsequent training sessions. In four sessions in each mouse after learning had occurred (d'>1), we introduced trials with various durations of whisker pad stimulation (1, 5, 10, 20, 40, or 100 ms) with the same total stimulus probability. Mice were rewarded with water for licking within the 2 s response window, as in standard training sessions. Trials with stimuli 20–100 ms duration were included in learning curves. In separate sessions, visual catch trials were introduced by delivering 460 nm light through a second optical fiber either in front of the head (so that light reached the eye). Catch trials were randomly interleaved (10% stimulus probability) with whisker pad optical stimulation trials; if mice licked during a catch trial, FA punishment was delivered. The second optical fiber was also used in separate behavior sessions to stimulate S1 with 460 nm light (10% stimulus probability). In this case, the fiber was placed <1 mm above the cranial window in a location corresponding to previously mapped S1. Licking responses within the 2 s response window after S1 stimulation were rewarded with water, as for peripheral stimulation.

## Histology

Two Emx1-Cre;Ai27D mice were deeply anesthetized and transcardially perfused with phosphate buffered saline (PBS) followed by 4% paraformaldehyde. A commercially available depilatory was applied to the anterior facial region to remove the overlying fur and most of the whiskers protruding from the skin. Whiskers not removed by the depilatory were trimmed close to the skin. The mystacial pad and underlying tissues were dissected and stored overnight in 4% paraformaldehyde at 4°C and then in 30% sucrose in PBS at 4°C until the tissue block had sunk. The tissue was then sectioned at 50 µm with a Leica CM1520 cryostat in either the coronal or transverse plane. Sections were mounted on slides and coverslipped with a glycerol-based mounting medium (KPL Inc., Gaithersburg, MD). Fluorescent micrographs in *Figure 1* were obtained using an Olympus IX51 microscope. Images used for montages were obtained with a 4x objective. Tissue from PV-Cre;Ai27 mice in *Figure 1—figure supplement 2* was processes similarly, except images were acquired along with Emx1-Cre;Ai27D tissue using an Olympus FluoView FV1000 confocal microscope and 20x objective.

Images for quantification were obtained using the same focus and illumination conditions for each set of measurements of the external extrinsic muscle, intrinsic muscle, and internal extrinsic muscle within a single follicle using a 10x objective. Intensities were measured from four follicles in transverse sections of the mystacial pad. Between follicles the focus and illumination conditions were adjusted to obtain optimal images. Only the field of view was adjusted to obtain images of the three

muscle types from within each follicle. To quantify ChR2/tdTomato expression in whisker musculature, fluorescence intensity was measured from three 50 x 50 pixel regions of interest (32.5 x 32.5 μm) placed over the muscle contained in 1392 x 1040 pixel 16-bit images using ImageJ (http://imagej.nih.gov/ij/). The intensity measured from each region of interest was then averaged to obtain a single intensity value for each image corresponding to a single intensity value for each muscle within the external to internal span of a follicle. A red look up table was applied to the images and levels were adjusted for display. Levels were adjusted equally for each panel in *Figure 1C3*.

### Statistics

Values are presented as mean ± SEM, unless otherwise noted. The number of subjects was chosen based on similarity to other in vivo studies and was not predetermined during design of the study. Statistical tests were carried out using Origin Pro or SAS. Student's two-tailed paired tests and one-way repeated measures ANOVA followed by paired contrasts were used for parametric data. Significance was measured at the level of $p < 0.05$.

## Acknowledgements

We thank Dr. Clay Lacefield and Elias Wang for assistance with Arduino, Dr. Qian Cai for assistance with histological processing, Dr. Kelvin Kwan for assistance with epifluorescence microscopy, and Drs. Huaye Zhang and Mladen-Roko Rasin for assistance with confocal microscopy.

## Additional information

### Funding

| Funder | Grant reference number | Author |
| --- | --- | --- |
| Whitehall Foundation | 2013-12-76 | David J Margolis |
| Brain and Behavior Research Foundation | NARSAD Young Investigator Award | David J Margolis |
| Rutgers Charles and Johanna Busch Biomedical Research Fund | N/A | David J Margolis |
| National Institutes of Health | R01NS094450 | David J Margolis |

The funders had no role in study design, data collection and interpretation, or the decision to submit the work for publication.

### Author contributions

SP, Conceived and designed the study, Performed whisker tracking and behavior experiments, Performed electrophysiology experiments, Performed data analysis, Wrote the paper with input from all authors; AB, Conceived and designed the study, Performed whisker tracking and behavior experiments, Performed histological analysis; CRL, Performed histology and histological analysis; DJM, Conceived and designed the study, Wrote the paper with input from all authors, Analysis and interpretation of data

### Author ORCIDs

David J Margolis, http://orcid.org/0000-0002-2678-4216

### Ethics

Animal experimentation: All procedures were approved by Rutgers University Institutional Animal Care and Use Committee (IACUC; protocol 13-033).

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

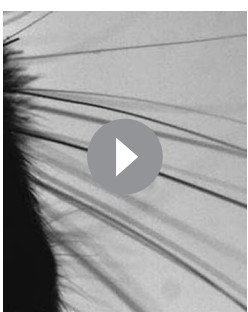

**Video 1.** Whisker movements in response to a light pulse at whisker pad position (0, 0), as in *Figure 1—figure supplement 1*. The blue square at the upper right indicates the timing of optogenetic stimulation (20 ms, 460 nm). The original sampling rate of 500 frames per second was slowed to 25 frames per second for display.

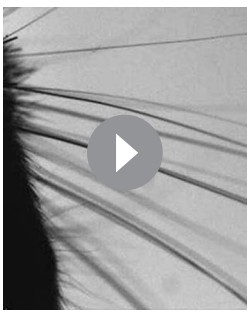

**Video 2.** Whisker movements in response to a light pulse at whisker pad position (-3, -2), as in *Figure 1—figure supplement 1*. The blue square at the upper right indicates the timing of optogenetic stimulation (20 ms, 460 nm). The original sampling rate of 500 frames per second was slowed to 25 frames per second for display.

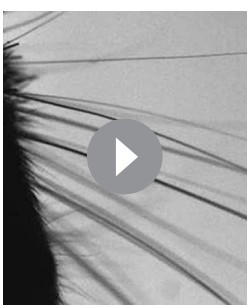

**Video 3.** Whisker movements in response to a light pulse at whisker pad position (-2, -2), as in *Figure 1—figure supplement 1*. The blue square at the upper right indicates the timing of optogenetic stimulation (20 ms, 460 nm). The original sampling rate of 500 frames per second was slowed to 25 frames per second for display.

