## [Decision Letter]

Thank you for submitting your work entitled "Peripheral optogenetic stimulation induces whisker movement and sensory perception in head-fixed mice" for consideration by *eLife*. Your article has been favorably evaluated by Gary Westbrook as the Senior editor and three reviewers, including Mitra Hartmann and Sacha Nelson, who is a member of our Board of Reviewing Editors. The reviewers have discussed the reviews with one another and the Reviewing Editor has drafted this decision to help you prepare a revised submission.

Summary:

The authors discovered that a commonly used driver strain mated to a mouse conditionally expressing ChR2 expresses in muscles that mediate whisking behavior. The authors realized that this could be used to investigate the reafferent sensory input that accompanies this form of active sensation. The whisker motions evoked optogenetically are shown to be perceptible and to generate signals that reach S1. The stimulation space is explored well and the work offers a guide for how people may use this technique in the future. The work will be a useful tool to add to the study of the vibrisso-trigeminal system.

Essential revisions:

1) The authors need to provide additional evidence that ChR2 is absent from the sensory neurons. One of the reviewers notes the presence of some apparent fluorescence near the shaft of the whisker. If expression is absent in the sensory afferents, this could be demonstrated with images of the IO nerve and/or ganglion. If on the other hand, there is some expression in the sensory nerve but the authors think they are selectively stimulating the muscles and/or facial motor nerve, they would have to devise some way of showing this convincingly, perhaps by showing that S1 does not respond to stimulation after muscle paralysis. Expression in the motorneurons is less problematic for the whole interpretation, but the manuscript should then make very clear that the present experiments cannot distinguish between motor nerve activation and muscle activation.

2) The paper should be identified as a "Tools and Resources" paper because the major thrust is to enable further studies.

3) The spatial extent of the activation should be more clearly described and analyzed.

A) The manuscript states that between 3 – 5 whiskers move each time but it is not clear whether this varies with stimulation intensity or location. It is not possible to tell from Figure 1 because the origin is never defined and does not state which whisker's (or whiskers') amplitude(s) is/are represented. Which whiskers are moving as different regions of the pad are stimulated?

B) If data are available, it would be highly informative to create supplementary figures for Figure 1 that represent each whisker's response to stimulation individually, placing the origin at each whisker and then moving to each of its neighbors in turn. How similar is the response of each of the whiskers?

For all of the plots related to Figure 1, the authors should use a colormap that diverges from zero. Results should state whether there was co-contraction during the 0 regions, or no contraction at all.

4) The adaptation index confounds the number of pulses with the frequency of pulses. A graph could be added to Figure 2 that computes the adaptation ratio after the same number of pulses instead of the same duration of stimulus.

---

## [Author Response]

*Essential revisions:*

1) The authors need to provide additional evidence that ChR2 is absent from the sensory neurons. One of the reviewers notes the presence of some apparent fluorescence near the shaft of the whisker. If expression is absent in the sensory afferents, this could be demonstrated with images of the IO nerve and/or ganglion. If on the other hand, there is some expression in the sensory nerve but the authors think they are selectively stimulating the muscles and/or facial motor nerve, they would have to devise some way of showing this convincingly, perhaps by showing that S1 does not respond to stimulation after muscle paralysis. Expression in the motorneurons is less problematic for the whole interpretation, but the manuscript should then make very clear that the present experiments cannot distinguish between motor nerve activation and muscle activation.

We performed additional analysis of histological sections at higher resolution to examine potential expression of ChR2/tdTomato in infraorbital nerve of Emx1-Cre;Ai27D mice. We found no evidence of expression in nerve fibers at the follicle. To rule out that this negative result was due to technical factors, we performed the same histological analysis in parvalbumin (PV)-Cre;Ai27D mice as a positive control. Previous work has shown that PV is a marker for Merkel endings of the IO nerve that innervate vibrissa follicles (Sakurai et al., 2013, The Organization of Submodality-Specific Touch Afferent Inputs in the Vibrissa Column, Cell Reports 5:87-98). Consistent with this, we found clear evidence for ChR2/tdTomato expression in PV-Cre;Ai27D follicular nerve axons and nerve endings along the shaft of the whisker, suggesting that we would have observed expression in Emx1-Cre;Ai27D mice if it were present. A comparison of sensory nerve ChR2/tdTomato expression in Emx and PV mice is now included in Figure 1—figure supplement 2. Although a full characterization of optogenetic stimulation effects in PV-Cre;Ai27D mice was beyond the scope of the present study, we hope to pursue this in future studies of Cre driver lines that target ChR2 expression to different components of vibrissal muscles/nerves. The methods we establish in the present study will form the foundation for such future investigations.

It should be noted that the histological results in Emx1-Cre;Ai27D mice, above, are consistent with the electrophysiological data in Figure 3 showing longer latency cortical LFP responses to optogenetic stimulation compared to mechanical whisker deflection. If ChR2 were expressed in follicular nerve endings, optogenetic response latencies would likely be as short as those evoked by mechanical whisker deflections. Instead, the observed longer latency responses to optogenetic stimulation fit the time scale of the evoked whisker movement, consistent with a reafferent rather than direct afferent origin of evoked sensory responses. Thus, our histological and functional data are consistent with the lack of ChR2 in sensory nerve in Emx-Cre;Ai27D mice, a point we clarified in the Results (subsection “Behavioral report of peripheral optogenetic stimulation”). ChR2 expression appears to be localized to extrinsic and intrinsic whisker pad muscles, but largely or completely absent from IO nerve.

2) The paper should be identified as a "Tools and Resources" paper because the major thrust is to enable further studies.

We concur with this classification.

*3) The spatial extent of the activation should be more clearly described and analyzed.*

A) The manuscript states that between 3 – 5 whiskers move each time but it is not clear whether this varies with stimulation intensity or location. It is not possible to tell from Figure 1 because the origin is never defined and does not state which whisker's (or whiskers') amplitude(s) is/are represented. Which whiskers are moving as different regions of the pad are stimulated?

We have updated the figure legend related to Figure 1 to clarify the location stimulated and the identity of the tracked whisker. We also performed further analysis of multiple whiskers in Figure 1—figure supplement 1 to address these important issues. Stimulation of a rostral region of the whisker pad The origin in Figure 1 (also diagramed as the position of optical stimulation in Figure 1) was defined at a rostral location that evoked reliable whisker protractions. We focused on stimulation of this location of the whisker pad throughout the rest of the study. Figure 1—figure supplement 1 now also addresses how different whiskers move as different regions of the pad are stimulated. We did this by plotting colormaps for multiple whiskers, as suggested below, and by plotting time courses of multiple whiskers for select stimulus locations (see next response for more details).

B) If data are available, it would be highly informative to create supplementary figures for Figure 1 that represent each whisker's response to stimulation individually, placing the origin at each whisker and then moving to each of its neighbors in turn. How similar is the response of each of the whiskers?

We included 5 additional color maps in Figure 1—figure supplement 1 (related to Figure 1) that represent the movements elicited by optogenetic stimulation of different regions of the whisker pad for each of 6 different whiskers. The color maps show a similar overall organization, including a rostral protraction area and a caudal-inferior retraction area. We chose to keep a common origin for each colormap, but similarities and differences for each whisker are clearly apparent. This organization is consistent with recent work by Haidarliu et al. (2015) on the structure of extrinsic musculature, which we now point out in the Discussion. We also performed a complementary analysis by plotting the evoked movement time courses for all tracked whiskers at a given stimulus location, which showed additional interesting features. Some sites, such as the rostral protraction area, elicited protractions that were relatively uniform across several whiskers. Other stimulation sites elicited more complex movements (usually of smaller amplitude), including a mixture of protraction and retraction from different whiskers. Although not the main focus of the paper, the phenomenon is now better documented in Figure 1—figure supplement 1, and the analysis better conveys the richness of movements that can be evoked by optogenetic whisker pad stimulation in Emx1-Cre;Ai27D mice.

For all of the plots related to Figure 1, the authors should use a colormap that diverges from zero. Results should state whether there was co-contraction during the 0 regions, or no contraction at all.

The color maps in Figure 1 and Figure 1—figure supplement 1 now diverge symmetrically from zero and use a different look up table to better visualize retractions and protractions. Because individual whiskers showed protractions or retractions depending on stimulus location, it is possible that stimulation at some locations elicits co-contractions that result in no net movement. No muscle activation would likely only result from stimulation delivered to the far edges of the whisker pad, but this was beyond the scale that we investigated. We added further description of these issues in the Figure 1—figure supplement 1 figure legend. In future work, we hope to achieve selective expression of ChR2 in different muscles or muscle types to further investigate this issue.

*4) The adaptation index confounds the number of pulses with the frequency of pulses. A graph could be added to Figure 2 that computes the adaptation ratio after the same number of pulses instead of the same duration of stimulus.*

We have added this graph to Figure 2 as a separate curve, and agree that this helps to separate pulse number from frequency, which more fully documents the adaptation effects. It should be noted that we discovered and corrected an error in the original analysis when reanalyzing data from one mouse, and that is why the graph for adaption index based on frequency appears different from the curve in our original submission.